# Mast Cells Limit Ear Swelling Independently of the Chymase Mouse Mast Cell Protease 4 in an MC903-Induced Atopic Dermatitis-Like Mouse Model

**DOI:** 10.3390/ijms21176311

**Published:** 2020-08-31

**Authors:** Sofie Svanberg, Zhiqiang Li, Pontus Öhlund, Ananya Roy, Magnus Åbrink

**Affiliations:** 1Evidensia Djurkliniken Öjebyn, Öjagatan 81, 94331 Öjebyn, Sweden; sofie.svanberg@evidensia.se; 2Section of Immunology, Department of Biomedical Sciences and Veterinary Public Health, Swedish University of Agricultural Sciences, VHC, Box 7028, 75007 Uppsala, Sweden; lzqforever@hotmail.com; 3Department of Immunology, School of Basic Medical Sciences, Guizhou Medical University, 550025 Guiyang, China; 4Section of Virology, Department of Biomedical Sciences and Veterinary Public Health, Swedish University of Agricultural Sciences, VHC, Box 7028, 75007 Uppsala, Sweden; pontus.ohlund@slu.se; 5Department of Immunology, Genetics and Pathology, Rudbeck Laboratory, Uppsala University, 75185 Uppsala, Sweden; ananya.roy@igp.uu.se

**Keywords:** mouse mast cell chymase 4 (mMCP-4), knockout, MC903, atopic dermatitis, IL-33, thymic stromal lymphopoietin

## Abstract

Atopic dermatitis (AD) is a complex, often lifelong allergic disease with severe pruritus affecting around 10% of both humans and dogs. To investigate the role of mast cells (MCs) and MC-specific proteases on the immunopathogenesis of AD, a vitamin D_3_-analog (MC903) was used to induce clinical AD-like symptoms in c-kit-dependent MC-deficient Wsh^−/−^ and the MC protease-deficient mMCP-4^−/−^, mMCP-6^−/−^, and CPA3^−/−^ mouse strains. MC903-treatment on the ear lobe increased clinical scores and ear-thickening, along with increased MC and granulocyte infiltration and activity, as well as increased levels of interleukin 33 (IL-33) locally and thymic stromal lymphopoietin (TSLP) both locally and systemically. The MC-deficient Wsh^−/−^ mice showed significantly increased clinical score and ear thickening albeit having lower ear tissue levels of IL-33 and TSLP as well as lower serum levels of TSLP as compared to the WT mice. In contrast, although having significantly increased IL-33 ear tissue levels the chymase-deficient mMCP-4^−/−^ mice showed similar clinical score, ear thickening, and TSLP levels in ear tissue and serum as the WT mice, whereas mMCP-6 and CPA3 -deficient mice showed a slightly reduced ear thickening and granulocyte infiltration. Our results suggest that MCs promote and control the level of MC903-induced AD-like inflammation.

## 1. Introduction

Atopic dermatitis (AD) develops after inflammatory responses against common environmental antigens (Ag) and is a lifelong disease with approximately 10 to 15% prevalence in both children and dogs [1,2]. Pruritus, the hallmark symptom of AD, causes a substantial impact on the quality of life for the dog patients, as well as the dog owners, and affected humans ([3,4] and references therein). The pruritus is usually followed by typical lesions, such as erythematous macules and flaky skin with comparable symptoms in humans and dogs [1,4,5]. These early lesions aggravate into chronic lesions with hyperpigmentation and lichenification due to self-trauma, and hyperplasia of the epidermis and corresponding thickening of the skin is a typical histological finding in the skin lesions of AD patients [6,7,8].

Mast cells (MCs) are present in most organs and abundant in tissues in contact with the surrounding environment, i.e., the skin, intestine, and mucous membranes [9,10]. MCs are situated close to nerves as well as lymphatic and blood vessels [11], and this strategic placement and the myriad of inflammatory mediators that MCs produce, store and secrete, gives MCs the capability to immediately influence the surrounding tissue and, to attract and regulate other inflammatory cells in the affected tissue [12,13]. Thus, MCs can act directly against environmental allergens and can play a key role in driving the immune response against a not so harmful Ag, e.g., house dust mite allergens, which causes the allergic reactions that develop into AD in both canines and humans [14,15].

MCs from humans, dogs, and mice share many morphological and functional characteristics. Hence the use of experimental mouse models to increase the understanding of MC-dependent immunologic disorders, such as AD, is pertinent. The chymase mouse mast cell protease (mMCP)-4, the tryptase mMCP-6 and the carboxypeptidase A3 (CPA3) are the predominantly expressed proteases in the mouse connective tissue MCs [16]. Importantly, the mouse chymase mMCP-4 share a similar expression pattern and tissue distribution with the human and canine MC chymases, as well as identical substrate cleavage specificity [17]. 

To study the role of MCs and mast cell proteases in AD we induced an AD-like inflammation by topical application of the low-calcemic vitamin D_3_-analog MC903 on the ear lobes of c-kit dependent MC-deficient (Wsh^−/−^) mice [18], or in mice lacking the MC-specific chymase (mMCP-4^−/−^) [19], or the MC tryptase (mMCP-6^−/−^) [20], or the MC carboxypeptidase A3 (CPA3^−/−^) [21]. Based on previous studies ([22] and refs therein), we hypothesized that treatment with a relatively low dose of MC903 would successfully induce an AD-like model and increase the inflammatory parameters analyzed, e.g., clinical scoring, migration of inflammatory cells, and inflammatory mediators. A functional MC driven AD-like model was established and we here show that the MC-mediated inflammation can be modulated by the MC-specific proteases. Interleukin (IL)-33 and thymic stromal lymphopoietin protein (TSLP), two critical regulators of AD pathogenesis were c-kit-dependently up-regulated in the MC903 induced AD-like model. The chymase mMCP-4 showed no major regulatory effect on the TSLP levels. In contrast, the IL-33 levels were significantly increased in mMCP-4-deficient ear tissue, suggesting an important role for chymase in the regulation of bioavailable IL-33 levels. 

## 2. Results

### 2.1. MC903-Treatment Significantly Increase the Clinical Score and Ear Thickness in c-Kit Dependent Mast Cell-Deficient Mice, Independently of the Mast Cell-Specific Chymase mMCP-4 

Daily topical application with MC903 induced redness, as the first sign of inflammation, after 5–8 days in the Wsh mouse strain. During the first week, only a minor increase in ear thickness was seen, which developed rapidly during the last days of the trial. At endpoint, i.e., on day 14 of treatment, these changes had developed into clearly visible thickening and dryness of the skin of the MC903-treated mice (Appendix A) and all MC903-treated ears had at least a clinical score of 1, which corresponds to mild redness and swelling. No visible changes were seen in any of the vehicle-treated ears (clinical scoring = 0), suggesting that it was MC903 that caused the significant redness and thickening of the ear and not the ethanol-treatment. The c-kit dependent MC-deficient Wsh^−/−^ mice developed a significantly higher clinical score (Figure 1a) and ear thickness (Figure 1b), as compared to the Wsh^+/−^ littermate mice. The highest increase in one MC903-treated ear from day 1 to day 14 was 0.57 mm (in a Wsh^−/−^ mice) and the lowest increase was 0.11 mm (measured in two Wsh^+/−^ mice). Histological analysis of H&E and toluidine blue stained ear sections showed that the thickening in MC903-treated ears seen in vivo corresponded to a thickened epidermis and ample leukocyte recruitment in the dermis (Appendix A). The epidermis was thicker in MC903-treated ears from Wsh^−/−^ than in ears from Wsh^+/−^ mice. Furthermore, an almost complete absence of MCs in Wsh^−/−^ mice (as expected) and several degranulated MCs in Wsh^+/−^ mice were observed.

The clinical score and ear thickness in mMCP-4^+/+,+/−^ and mMCP-4^−/−^ mice was significantly increased by the MC903-treatment, from day 7–8, as compared to both vehicle-treated ears and the non-treated group (Figure 1c,d). However, the lack of chymase mMCP-4 did not significantly change the clinical score (Figure 1c) or the ear thickening (Figure 1d) as compared to the wild type (WT) and heterozygous littermate mice. No apparent morphological difference or difference in leukocyte recruitment was seen between the MC903-treated mMCP-4^+/+^ and mMCP-4^−/−^ mice (Appendix A), with similar infiltrating mast cell and granulocyte numbers (Appendix A). 

### 2.2. MC903 Induces Significantly Increased MC Numbers and Activity in the Ear Tissue 

We next investigated if the lack of chymase affected MC infiltration and activation following MC903-treatment. In MC903-treated ears, as compared to vehicle-treated ears, significantly higher numbers of MCs (Figure 2a), increased percentage of degranulated and thus activated MCs (Figure 2b), as well as higher tryptase activity (Figure 2c) were found, but with no significant difference between the mMCP-4^+/+,+/−^ and mMCP-4^−/−^ mice. Visual comparison of auricular lymph nodes collected at endpoint day 14 clearly showed an enlarged left lymph node draining the MC903-treated ears compared to the lymph nodes on the right side draining the vehicle-treated ears, both in the Wsh and the mMCP-4 mouse strains. Mast cell counts from the auricular lymph node cell suspensions of mMCP-4^+/+,+/−^ and mMCP-4^−/−^ mice, showed significantly increased MC numbers in the MC903-treated mMCP-4^+/+,+/−^ ear draining lymph nodes, while mMCP-4^−/−^ showed only a trend, as compared with the vehicle-treated side. However, no significant difference was observed between the mMCP-4^+/+,+/−^ and mMCP-4^−/−^ mice (Figure 2d).

To investigate if MC903 directly cause activation and degranulation of MCs, peritoneal cell-derived mast cells (PCMCs) were stimulated with 1, 5, or 10 nmol of MC903 for 1 h and overnight. All concentrations of MC903 were found to induce activation and degranulation of the PCMCs, whereas the vehicle (ethanol) alone had little effect (Appendix A).

### 2.3. Topical Application of MC903 Induces a Significant Increase of Granulocytes and Neutrophil Elastase Activity in Ear Tissue

To further investigate the MC903-induced inflammation H&E stained ear tissue sections were evaluated for granulocyte infiltration. Compared to vehicle, MC903-treatment induced a significantly increased granulocyte infiltration in the Wsh^+/−^ and Wsh^−/−^ ears (Figure 3a) as well as in the mMCP-4^+/+,+/−^ and mMCP-4^−/−^ ears (Figure 3c). However, no significant difference in granulocyte infiltration was seen between the MC903-treated Wsh^+/−^ and Wsh^−/−^ mice or the mMCP-4^+/+,+/−^ and mMCP-4^−/−^ mice. To further assess the granulocyte function, we next evaluated the activity of neutrophil elastase (NE) in the ear tissue. Albeit only a low activity of NE was found in the ears, the NE activity was higher in MC903-treated Wsh^+/−^ and Wsh^−/−^ ears as well in MC903-treated mMCP-4^+/+,+/−^ ears compared to vehicle-treated ears from the same mice (Figure 3b,d), whereas MC903-treated mMCP-4^−/−^ ears only showed a trending increase of NE activity compared to the vehicle-treated mMCP-4^−/−^ ears (Figure 3d). The NE activity showed no significant difference between the mMCP-4^+/+,+/−^ and mMCP-4^−/−^ MC903-treated ears.

### 2.4. MC903 Treatment Increases the IL-33 Levels in Ear Tissue and TSLP in Ear Tissue and Serum

Thymic stromal lymphopoietin (TSLP) and IL-33 are two cytokines previously shown to be AD-associated [23,24,25,26]. Therefore, we next evaluated the impact of MCs and the MC specific chymase mMCP-4 on IL-33 and TSLP levels in the MC903-induced AD-like mouse model. In the MC903-treated ears, IL-33 levels were significantly increased as compared with vehicle-treated ears and non-treated control ears (Figure 4a,b). While the c-kit deficiency caused a non-significant reduction of IL-33 (Figure 4a), the chymase-deficiency caused significantly higher levels of IL-33 in the ear tissue as compared to the mMCP-4^+/+,+/−^ mice (Figure 4b).

When assessing TSLP at endpoint the MC903-treatment resulted in 15 to 50-fold increased levels both in the Wsh^+/−^ and Wsh^−/−^ ears, and in the mMCP-4^+/+,+/−^ and mMCP-4^−/−^ ears, as compared to the vehicle-treated ears (Figure 4c,d). Lack of c-kit signaling (Wsh^−/−^) resulted in lower concentrations of TSLP in the MC903-treated ears (Figure 4c) and serum (Figure 4e) day 14 as compared with the Wsh^+/−^ mice. In contrast, the lack of chymase caused no significant difference in the local or systemic levels of TSLP (Figure 4d,f). The MC-specific chymase mMCP-4 did not significantly affect the increased TSLP serum levels at day 7 (≤10 fold increase as compared with non-treated animals) or day 14 (≥3 fold increase) (Figure 4f). Note that the serum levels of TSLP day 3 post-treatment were not significantly higher than in non-treated mice (Figure 4f), suggesting that the low dose of MC903 induced a slower disease progression than what Li et al. reported for a high dose of MC903, for example [27]. This indicates a tentative influence of MCs on the clinical levels of TSLP that however is not modulated by the MC-specific chymase mMCP-4.

### 2.5. MC903-Treatment Induce Local Production of a Wide Range of Inflammatory Cytokines 

To further investigate the role of the mast cell chymase mMCP-4 in the induction of local cytokine levels following MC903-treatment, a qualitative cytokine array analysis was performed on the day 14 ear tissue lysates from the mMCP-4^+/+, +/−^ (*n* = 4 per genotype was pooled) and the mMCP-4^−/−^ (*n* = 4 was pooled) mice, which all showed a high level of TSLP expression. The results showed a massive production of inflammatory mediators (Figure 5). In addition to a prominent proinflammatory response, as shown by increased levels of e.g., CCL2, MIP-1 alpha and beta, IL-1, IL-6 TNF-α, several cytokines associated with Th2-type immune responses were indeed present at day 14 in this AD-model, e.g., IL-4, IL-5, and IL-13, while the Th1-associated cytokines IL-12 and IFN-γ showed no activity or was present only in very low levels, thus suggesting that topical application with MC903 induced predominantly a Th2-type cytokine response profile (Figure 5, upper panel).

### 2.6. The Tryptase mMCP-6, the Zinc Metalloprotease CPA3 and the Genetic Background of the Mice Affects Ear Thickness in the MC903 Induced AD-Like Mouse Model 

Seeing the relatively mild impact of the MC specific chymase mMCP-4 on the MC903-induced AD-like model we finally scored the ear thickness as well as MC and granulocyte infiltration in 12 to 18 weeks old C57BL/6J Taconic mice lacking the tryptase mMCP-6 or the carboxypeptidase A3 (CPA3). 

While the mMCP-6-deficient mice only showed a trend for reduced ear thickness at day 14 (Figure 6a), a small but significant difference in ear thickness at day 14 was observed and CPA3^−/−^ mice as compared with wild type littermate mice (Figure 6b). The MC903-treatment induced significantly increased MC counts in the ear tissues of WT, mMCP-6^−/−^ and CPA3^−/−^ mice as compared to the vehicle-treated ears (Figure 6c), with similar numbers of MCs as seen in the 6 to 9 weeks old mMCP-4^−/−^ mice (Figure 2a). The granulocyte infiltration into ear tissue was significantly lower in the mMCP-6^−/−^ and CPA3^−/−^ mice (Figure 6d), suggesting that tryptase and CPA3 act pro-inflammatory in this experimental AD mouse model. Finally, we compared the development of ear thickness in MC903-treated mice on the BALB/c Taconic genetic background with the development of ear thickness in C57BL/6J Taconic MC903-treated mice (Appendix A). The BALB/c mice, which is known to respond with a pronounced Th2-prone cytokine profile, showed at endpoint a significantly increased ear thickness as compared with the C57BL/6J mice. This suggests that the genetic background of the mice could affect the results and that the experimental AD-model may require backcrossing of the gene of interest to the “best” genetic background.

## 3. Discussion

In this study, we used topical application of 1 nmol MC903 on mice ears for 14 days to mimic AD. Various studies have reported the use of between 1 to 4 nmol of MC903 for topical application on the ears, with higher doses causing a rather rapid and severe AD development ([22] and references therein). For example, Li et al. 2006, used 4 nmol MC903 over 17 days and the mice developed severe AD-like symptoms including pruritus [27]. However, several studies on experimental allergic models suggest that the effective impact of MCs possibly may be overruled by a too high dose of antigen [28,29,30,31,32,33]. We, therefore, decided to use a low dose of MC903 and applied daily 0.5 nmol MC903 dorsally and ventrally on the left ears of the mice, and only vehicle on the right ears for internal control. The low dose of MC903 induced inflammatory changes in all mouse strains studied, similar to those seen in human and canine AD-patients, thus confirming the use of a low dose of MC903 to create a functioning mouse model of AD. Macroscopically, MC903 induced changes mimicking a naturally occurring AD, with redness, dermal thickening, and flaky skin, which corresponded well to the early changes seen in humans and dogs [1,4,5]. If MC903 evoked pruritus, the most prominent symptom of human and canine AD was not evaluated in this study. However, towards the end of the MC903-treatment an increase in the scratching behavior, usually directly after the daily topical application, was observed. The study protocol was ended at day 14 before any mice showed severe clinical changes, why none of the later changes in AD, e.g., hyperpigmentation was documented. 

The c-kit-dependent MC-deficient Wsh^−/−^ mice displayed a significantly stronger reaction to MC903-treatment in both scoring and ear thickening, thus implicating a limiting role of MCs in the clinical development of AD. However, the chymase mMCP-4 cannot be accounted for behind the development of these increased clinical symptoms, as no significant clinical difference was found between mMCP-4^+/+,+/−^, and mMCP-4^−/−^ mice. Epidermal thickening with inflammatory cell infiltration to a varying degree was seen in all Wsh^+/−^, Wsh^−/−^, mMCP-4^+/+,+/−^ and mMCP-4^−/−^ MC903-treated ears, similar to the typical changes known to exist in naturally occurring AD [4,6]. 

MCs are known to alter homeostasis and enhance adhesion of neutrophils to the endothelium, which enables migration into the skin [15]. Compared to vehicle-treated ears the MC903-treatment increased the number of granulocytes in the ear tissues of all mouse strains studied. Interestingly, although the MC-deficient Wsh^−/−^ mice showed a higher clinical score the granulocyte counts and NE activity were similar to Wsh^+/−^ mice. Hence, the neutrophilia described for the Wsh^−/−^ mice [34] does not seem to affect the granulocyte/neutrophil numbers and NE activity in the ear tissue in the current AD-model and therefore does not intervene with our results. In the mMCP-6^−/−^ and the CPA3^−/−^ mice, a significantly reduced granulocyte infiltration was observed, which reflected in a small reduction of ear thickness in both strains. In contrast, the absence of the chymase mMCP-4 resulted in only a tendency for reduced granulocyte numbers and NE activity with no difference in the clinical score or ear thickness. Hence it is probable that chymase has no major role in granulocyte recruitment in the current MC903-induced AD-like mouse model. However, the genetic background of the mice could also be of importance [33]. We observed significantly increased ear thickening in the WT BALB/c Taconic mice as compared with WT C57BL/6J Taconic mice, thus suggesting that the Th2-prone response profile in the BALB/c mouse strain influences the severity of the MC903-induced AD. Thus, in a future perspective, it would be interesting to evaluate the mMCP-4-deficient mice on the BALB/c genetic background in the MC903-induced AD-like mouse model.

In the mMCP-4^+/+,+/−^ and mMCP-4^−/−^ C57BL/6J Taconic littermate mice tryptase activity increased in the MC903-treated ears, and MC numbers were increased in the auricular lymph nodes draining the MC903-treated ears, suggesting that MC903 can activate MCs and may cause migration of these cells into the lymph nodes. A notion which was confirmed by the activation of PCMCs by MC903 in vitro. The lack of the chymase mMCP-4 did not affect MC-activation significantly, neither was there any significant difference in MC numbers locally or in the draining lymph nodes suggesting that the lack of mMCP-4/chymase does not affect MC migration in the current AD-like model. 

MCs can contribute to the promotion of a Th2-dominated (type 2) inflammation in the tissue and the lymph nodes, through the production of Th2-associated cytokines [35], and potentially proteolytic degradation of Th1-associated cytokines, such as TNF-α [36]. Piliponsky and co-workers showed that chymase/mMCP-4 may degrade TNF-α during peritoneal bacterial infections to reduce and control the level of the early inflammatory response, at 6 h after the infection [36]. However, we found similar levels of TNF-α on the cytokine arrays of the day 14 MC903-induced ear tissue lysates of the mMCP-4^+/+, +/−^ and mMCP-4^−/−^ mice. This may either suggest that TNF-α levels are not regulated by chymase in the ear tissue in the MC903 induced AD-like model, or that the regulatory role on TNF-α by chymase occurs earlier during the induction phase of the skin inflammation. Another study recently showed that the human MC chymase in fact could degrade very few cytokines of the 50 cytokines analyzed [37], i.e., only IL-15, IL-18, IL-33 showed almost complete degradation whereas CTFG, Flt3L, IL-3, IL-6, IL-13, and TSLP showed minor degradation. Interestingly, the human chymase did not significantly degrade human TNF-α, suggesting minor differences in the potential target structures of TNF-α between mouse and human. Thus, the potential role of chymase/mMCP-4 in the regulation, via degradation or activation, of TNF-α and other cytokines induced in the MC903-induced AD-like mouse model (Figure 5) would be interesting to follow up. Furthermore, previous studies showed that MC903 treatment promotes a Th2 immune response with significantly increased levels of IgE [27,38], levels which however seems to be independent on c-kit signaling [33,39,40]. 

Our data reveals AD-like inflammatory changes upon MC903-treatment, for example, the presence of Th2-associated cytokines such as IL-4, IL-5, and IL-13 and only very low levels of Th1-associated cytokines. An increase of the AD-associated cytokines TSLP and IL-33 was also observed upon MC903 application although the levels of IL-33 and TSLP in the c-kit dependent MC-deficient Wsh^−/−^ mice was lower than in WT mice. This data is in line with the observation by Hueber et al. 2011 where they demonstrate that the response to IL-33 and subsequent inflammatory effect seems to be impeded in MC-deficient mice [41]. Diverse modes of regulation of IL-33 and Th2-mediated responses have been reported in the pathophysiology of AD. For example, Type 2 innate lymphoid cells (ILC2), activated by IL-33 via the ST2 receptor in an MC903-induced AD BALB/c mouse model, was shown to be important for the production of type 2 cytokines (Th2), such as IL-5 and IL-13. Deletion of ILC2 cells or the ST2 receptor in BALB/c mice significantly reduced ear swelling, whereas the deletion of the TSLP receptor showed reduced impact on ILC2 numbers and ear swelling [42]. In an MC903-induced (2 nmol) AD mouse (C57BL/6) model blocking of ST2 signaling via MyD88 reduced ear thickness and the level of Th2 cytokines [43]. Furthermore, the deletion of JNK1, a kinase phosphorylating the transcription factor c-Jun that regulates the Th2 cytokine profile, reduced the AD-like symptoms in mice on the C57BL/6J background [38]. Ectopic expression of IL-33 in skin keratinocytes have been shown to increase spontaneous AD, manifested with increased Th2 cytokine profile, the activity of ILC2 and MCs, and pruritus [44]. In addition, TSLP activates neurons and promotes itch [45], whereas IL-33 promotes the production of IL-31 from leukocytes, where IL-31 is a strong inducer of itch [26]. Our results are in line with these reports, where we see an increase in IL-33 and TSLP as well as the Th2 cytokine profile in our MC903-induced AD-like mouse model. However, the importance of IL-33 and ST2 was recently challenged in a high dose MC903-induced (4 nmol) AD model in C57BL/6J Taconic mice, where even the double knockout deleting both IL-33 and ST2 showed similar development of ear thickness, leukocyte infiltration and transcription of IL-4 and IL-13, as the WT mice [46]. The high dose of MC903 (4 nmol) used in the study by Pietka et al. 2020 [46] may explain the differences to the results presented by Salimi et al. 2013 [42] and Li et al. 2017 [43], which used a lower dose of MC903.

Interestingly, our results indicate a role for MCs in driving the increase of TSLP and IL-33 but, at the same time suggesting that these cytokines are not drivers of the scored clinical AD-like symptoms, since the c-kit dependent MC-deficient Wsh^−/−^ mice had lower levels of TSLP and IL-33 but higher clinical scoring. Likely, a significant part of MCs contribution to the increase of TSLP and IL-33 occurs indirectly, through the production of mediators like IL-4 and IL-13 that enhance TSLP and IL-33 -expression in other cells, e.g., epithelial cells and keratinocytes [9]. In contrast, the TSLP protein levels were similar in mice lacking mMCP-4 as compared to the WT and heterozygous mice, suggesting that chymase does not affect TSLP expression in MC903-induced AD. Interestingly, the levels of TSLP were lower in the current study as compared with the study by Li et al. 2006 [27], which used 4 nmol MC903 to rapidly induce very high mRNA levels of TSLP in the ear tissue (from day 3) and significant TSLP protein levels in serum from day 2, reaching ≈7 ng/mL at day 4. In the current study, the use of 1 nmol MC903 induced a slower AD development which manifested by a >7-fold lower level of TSLP in serum (≈1 ng/mL) day 7 (Figure 4f) as compared with the day 4 levels in the study by Li et al. 2006 [27]. They also showed that mRNA levels of TSLP remained high in ear tissue at day 17 in their model. In addition, IL-33 was also increased in the mMCP-4^+/+, +/−^ and mMCP-4^−/−^ mice after MC903-treatment and, interestingly, the chymase-deficient mice had significantly higher levels of IL-33 than WT mice implying a limiting role for chymase on the bioavailable levels of IL-33, likely via degradation or cleavage of IL-33 [47,48]. However, as the clinical parameters in the mice lacking mMCP-4 were not significantly different from the mMCP-4-competent mice, our result suggests that the regulation of the IL-33 levels by chymase is not a driving factor in the current AD-model.

In summary, we show a limiting role of c-kit signaling-dependent resident cells, i.e., MCs, in the clinical development of the MC903-induced AD-like symptoms. In the Wsh^−/−^ mice several parameters such as levels of TSLP and IL-33 were lowered either locally or systemically, or both, suggesting that the MC903-induced AD-model is MC-dependent. In contrast, the chymase mMCP-4 only exhibited a limitation of the local levels of IL-33. TSLP, another important cytokine in the MC903-induced AD-model, was relatively unaffected by the presence or absence of mMCP-4/chymase. Our data demonstrate small and varied effects of three major MC proteases in eliciting clinical symptoms, granulocyte infiltration, and overall cytokine responses although neither of the proteases though showed any significant overall effect in limiting AD. Thus, the absence of MCs and not deletions of single MC proteases significantly affected the clinical and systemic symptoms in the 1 nmol MC903-induced AD-like model. To further address the potential functional role of the MC proteases, the triple protease knockout lacking the three major proteases, i.e., mMCP-4, mMCP-6 and CPA3 [49] in comparison to the single knockout strains, would be highly interesting to evaluate in the MC903-induced AD-model, with varying doses of 1 to 4 nmol of MC903 and in mice on both the C57BL/6J and the BALB/c genetic backgrounds.

## 4. Materials and Methods 

### 4.1. Experimental Animals

The methods and mouse strains used in the MC903-induced experimental AD-like model described in this study was approved by the Uppsala District Court Ethical Committee for Use of Experimental Animals (permit C66/13 approved 2013-06-26, and permit C140/15 approved 2015-12-18). All knockout mouse strains used in this study were congenic on the C57Bl/6J Taconic genetic background: i.e., the Wsh^−/−^ mouse strain, with nearly a total loss of MCs due to a genomic inversion of the c-kit locus [18]; the mouse mast cell protease (mMCP)-4^−/−^ mouse strain, only lacking the MC-specific protease chymase [19]; the tryptase-deficient mMCP-6^−/−^ mouse strain [20]; the carboxypeptidase A3-deficient CPA3^−/−^ mouse strain, which however also lack the mMCP-5 protease with elastase-like activity [21]. 

Six to nine weeks old Wsh (+/− *n* = 7; −/− *n* = 7) and mMCP-4 (+/+ *n* = 6; +/− *n* = 19; −/− *n* = 11) female and male littermate mice were used in the MC903-induced AD-like model. The AD-like symptoms in the littermate mMCP-4^−/−^, mMCP-4^+/−^, and WT mice were scored in a blinded fashion, i.e., the genotype of the 36 treated mice and five control mice was determined by polymerase chain reaction (PCR) after the terminal endpoint of the experiment. Five Wsh^+/−^, five Wsh^−/−^, and five WT littermate mice were also included as untreated controls. In addition, to address the potential role of other major MC proteases in AD twelve to 18 weeks old mMCP-6^−/−^ (n = 12), CPA3^−/−^ (n = 6), and wild type (WT) (*n* = 6) mice were also challenged with MC903 and clinically scored in a blinded fashion. To compare differences due to the genetic background 6 WT twelve to 18 weeks old mice on the BALB/c genetic background were also included in the study. All of the animals were kept in a controlled environment in individually ventilated cages, cared for by trained staff, and fed with R3 Pellets (Lantmännen, Stockholm, Sweden). 

### 4.2. Induction of Atopic Dermatitis on Ears by MC903 and Clinical Scoring

An AD-like disease was induced by the topical application of a low-calcemic vitamin D_3_-analog, MC903. This chemical substance is also known as calcipotriol (Dovonex), used for the treatment of psoriasis. Several studies have been performed to investigate the functions of MC903 and its potential as an AD-inducer [22,27,50] and refs therein). In the present study, 1 nmol MC903 in 20 μL 95% ethanol were topically applied on the left ear, 10 μL on each side of the ear. On the right ear, the same amount of 95% ethanol only was applied as an internal vehicle control. This procedure followed once a day for a total of 14 subsequent days. The development of AD-like changes was followed by photography, daily measuring of ear thickness using a digital engineer’s micrometer (Mitutoyo Corporation, Sakado, Japan) and by clinical scoring: 0 = no symptoms, 1 = mild redness/swelling, 2 = redness/swelling, 3 = redness/swelling/bleeding.

### 4.3. MC903 Induction of Peritoneal Cell-Derived Mast Cells

To verify that MCs can respond directly to MC903, peritoneal cell-derived mast cells (PCMCs) were derived by culturing peritoneal lavage cells for 3 to 4 weeks in conditioned media (as described in Malbec et al. 2007 [51]). PCMCs (10^6^ per mL) were seeded in 24 well plates and stimulated with 1 and 5 nmol MC903 diluted in 95% ethanol, for 1 h and overnight, and then collected on object glasses by cytospin (700 rpm, 5 min) before the activation status of the PCMCs were manually controlled using a light microscope (Nikon Eclipse E200, Tokyo, Japan). As a control, the PCMCs received only 95% ethanol, without MC903.

### 4.4. Sampling of Material from Mice

From the mMCP-4 mouse strain blood was obtained from the tail vein twice during the trial, on day 3 and day 7, in amounts up to 100 μL. To enhance peripheral blood flow, a heating lamp was placed above the cage for five minutes before blood sampling. At the end of the trial, mice were sacrificed using carbon dioxide (CO_2_). Thereafter ears, auricular lymph nodes, blood, and a piece of the tail (for genotyping) were collected. All of the collected materials were used for different analyses in this study.

### 4.5. Genotyping of mMCP-4^+/+^, mMCP-4^+/−^, and mMCP-4^−/−^ Mice

The genotype in the Wsh mouse strain is apparent in the phenotype since homozygous mice are white, heterozygous mice are black and white (with a white sash around the waistline) and WT mice are black. This phenomenon is caused by the inversion mutation in the c-kit gene that does not only inhibit MCs but also affects melanocytes [18]. Therefore, only mMCP-4^+/+^, mMCP-4^+/−^, and mMCP-4^−/−^ littermate mice on the C57Bl/6 background were genotyped. Genotyping was performed using a KAPA Mouse Genotyping kit (KAPA Biosystems, Wilmington, MA, USA) and with primers, as previously described [52].

### 4.6. Histopathology of Ears

To estimate microscopically visible inflammatory parameters, the histopathologic evaluation was performed on all mouse ears collected at the end of the trial. Ears were cut into two halves; one-half of the ears were soaked in 1 mL of 30% sucrose for 48 h before they were embedded in medium (CryoMount), and quick frozen on dry ice. Subsequently, the embedded tissues were sectioned in a Cryostat at 8 μm slices and transferred onto microscope slides (Superfrost^®^ plus, Thermo Scientific, Waltham, MA, USA). The sectioned tissues were fixed in 4% paraformaldehyde for 1 h, before staining with hematoxylin-eosin (H&E) or toluidine blue (TB) to study cellular infiltration. Slides were digitally captured for presentation using NIS-Elements Microscope Imaging Software (Nikon) in a Nikon Eclipse 90i microscope. 

### 4.7. Lymph Node Cell Counts

Left and right auricular lymph node (draining the ears) were collected and separately mixed with 600 μL phosphate-buffered saline (PBS) into a cell suspension. Auricular lymph node cells from all of the treated mice in the trial plus a lymph node collected from each of the 15 control animals (untreated) were analyzed. For Wsh animals, 100 μL lymph node cell suspensions, diluted 1:5 to 1:20 depending on cell density and containing 50′ to 100′ cells, were loaded into the funnel of the cytospin glass container. For the mMCP-4^+/+^, mMCP-4^+/−^, and mMCP-4^−/−^ animals, 200 μL lymph node cell suspensions diluted 1:30 for the vehicle side and 1:60 for MC903 side, were loaded into the funnel of the cytospin glass container and spun in a cytospin centrifuge for 5 min at 700 RPM.

All slides were subsequently stained with May–Grünwald–Giemsa solution. Thereafter, manual MC counts were performed on the whole surface containing cells, using a light microscope (Nikon Eclipse E200) at 400× magnification. All slides were blinded from individual markings during cell counting. In the results, lymph nodes were divided into groups by genotype (^+/+^ and ^+/−^ or ^−/−^), and by treatment (MC903 or vehicle).

### 4.8. Cell Counts on Ear Tissue Slides

Inflammatory cell counts in the ear tissue were made using light microscopy (Nikon Eclipse E200) at 400× magnification of the histological slides. Numbers of cells were manually counted in five to twelve randomly selected viewing fields at 400× magnification in each slide; results presented as individual means in genotype groups. All slides were blinded from individual markings during cell counting. Granulocytes were counted in slides stained with H&E and MCs were counted in slides stained with toluidine blue (TB).

### 4.9. Enzyme-Linked Immunosorbent Assays

Enzyme-linked immunosorbent assay (ELISA) was used for the quantitative analysis of TSLP and IL-33. TSLP was chosen because of its previously described connection with human and canine AD [53]. To analyze quantities of TSLP in serum and ear tissue, a commercial ELISA kit (eBioscience, San Diego, CA, USA), was used. For the preparation of protein suspension used in the ELISA, the other halves of the mouse ears were deep-frozen with liquid nitrogen and crushed into a fine powder. The powder was solved in 500 μL PBS with 1% Triton-X; divided into two Eppendorf tubes in which one of them, proteinase inhibitor (PI) was added to prevent enzymatic digestion (Complete™, Mini PI cocktail Roche, Basel, Schweiz). All tubes were held on ice for 2 h; centrifuged for 10 min at 13′ RPM, thereafter both supernatants and pellet were frozen in –20 °C. Ear suspension supernatants in duplicates and diluted 1:20 were used for measuring local TSLP-levels using mouse TSLP ELISA Ready-SET-Go! (eBioscience), according to instructions and with materials that followed the kit. Ear suspensions from 5 MC903-treated Wsh^+/−^, 6 Wsh^−/−^ mice, and 5 non-treated controls in both genotypes were analyzed, as well as 14 mMCP-4^+/+, +/−^ mice and 6 mMCP-4^−/−^ mice. The plates were read at a wavelength of 450 nm (A_450_) to produce optical density (OD) values, subsequently converted into concentrations using a standard curve. With the same kit, TSLP-levels were also analyzed in serum (diluted 1:7) obtained at day 3 and 7 as well as at the end of the trial.

The concentration of the cytokine IL-33 was analyzed in ear tissue suspensions because of its previously described association with AD as well as its potential as an in vivo substrate for chymase [47,48]. Ear suspension supernatants in duplicates, diluted 1:20, from ten WT/mMCP-4^+/−^ mice and ten mMCP-4^−/−^ KO mice were analyzed using a DuoSet mouse IL-33 ELISA Development kit (R&D Systems, Minneapolis, MN, USA). Manufacturer’s protocol was followed, using supplied materials following the kit. 

### 4.10. Cytokine Array

A commercial kit, Mouse Cytokine Array Panel A (R&D Systems, Minneapolis, MN, USA), containing duplicate spots of capture antibodies directed against 40 different inflammatory cytokines and chemokines, was used to detect the mediators that are locally present after induction of AD with MC903. Ear suspensions from twelve mice were chosen, due to their high TSLP concentration as a marker of inflammation. Four mice had WT genotype, four mice mMCP-4^+/−^ heterozygous genotype and four mice were of mMCP-4 KO genotype. Ear suspensions with PI, 50 μL from each mouse in one genotype, were pooled into three samples with 200 μL that were used on each membrane. These samples were analyzed using the array procedure according to instructions appurtenant in the kit, with kit-specific buffers and reagents. The detection of bound cytokines was mediated with paired antibodies. The secondary antibody was biotin-labeled and thereafter streptavidin/dye was added. The signal was read in the Odyssey Imaging system (LI-COR Biosciences, Lincoln, NE, USA). Results were calculated as the mean pixel density of the two spots, representing one cytokine or chemokine in the two genotypes. Since negative background values were removed in calculations, all cytokines with a visible bar in the figure were also present in the ear. The pooled 4 WT and 4 heterozygous ear tissue suspensions gave very similar mean pixel data on the array and were therefore pooled and compared against the mMCP-4^−/−^ genotype (see Figure 4). 

### 4.11. Enzymatic Assays

The activity of the MC protease tryptase was measured in duplicate ear tissue extract suspension samples from 7 mMCP-4^+/+, +/−^ and 8 mMCP-4^−/−^ mice, using the chromogenic substrate Chromogenix S-2288 (DiaPharma, West Chester, OH, USA). Per well, 10 µL of ear suspension supernatant was mixed with 90 µL H_2_O in a 96-well plate, which was read at 405 nm. Thereafter, 20 µL of the 1.8 mM enzyme-substrate (S-2288) was added, and the plate was read every 15 min until 2 h, after which one final overnight reading was done. Results were calculated as differences in OD-values per hour (A_405_). 

To measure the local activity of the neutrophil enzyme elastase in ears, as the second parameter of neutrophil activity complementing the granulocyte counts, ear suspension supernatants from 25 MC903-treated ears (whereof 11 KO mice) and 12 vehicle-treated mice (whereof 4 KO mice) were analyzed. 20 µL duplicate samples from each ear suspension were incubated in a 96-well plate at room temperature, together with 20 µL of enzyme-substrate (Suc-Ala-Ala-Pro-Val-pNA) in a final concentration of 1 mM. To control the salt concentration, a balanced buffer containing 150 mM NaCl 0,05%, 100 mM TrisHCl, 0,1% BSA and Tween-20 (pH 8.5) was added to a final volume of 200 µL per well. Duplicates with buffer only, and buffer with the substrate, were used as blanks. OD values (A_405_) were determined every 15 min for 2 h. One last reading was performed after incubation overnight. Elastase activity was calculated as the difference in OD-values per hour. 

### 4.12. Statistical Analysis

Figure panels were made in GraphPad Prism 8 and statistical analyses of differences between genotypes at experimental endpoints were performed with the Brown–Forsythe and Welch’s ANOVA tests, not assuming equal standard deviations between groups and without Dunnett T3 correction for multiple comparisons (i.e., unpaired *t*-test with Welch’s correction), unless otherwise specified in the figure legends. *p*-values < 0.05 were considered significant.

## Figures and Tables

**Figure 1 ijms-21-06311-f001:**
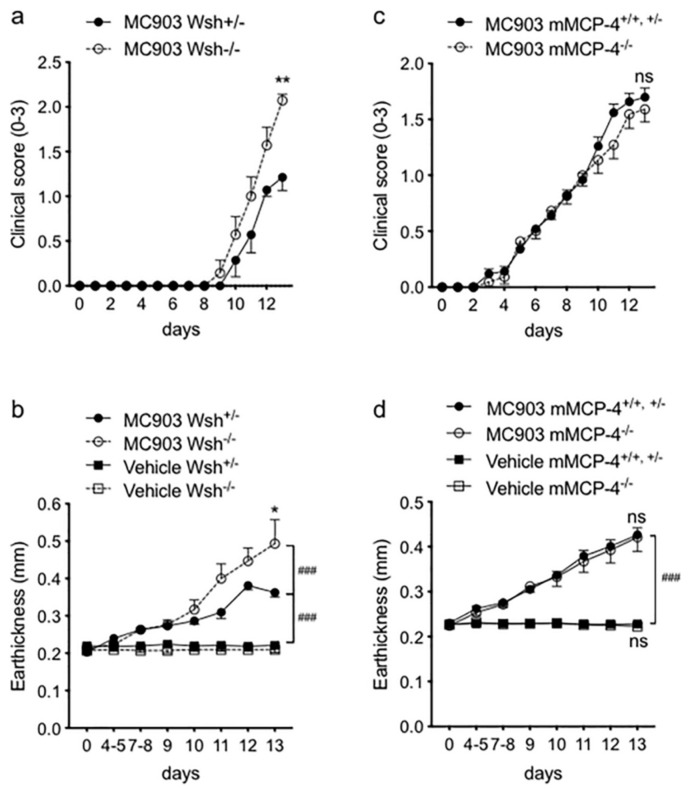
Mast cells affect clinical symptoms in ear tissue after the MC903 application. Significant increase in clinical scores (**a**) and ear thickness (**b**) in c-kit-dependent mast cell-deficient W^sh^^−/−^ mice compared to mast cell-competent W^sh+/−^ mice at day 14. In contrast, the mast cell-specific chymase mMCP-4 does not impact on the (**c**) clinical score and (**d**) ear thickness of MC903-induced atopic dermatitis. Age-matched 6–9 weeks old W^sh+/−^ (*n* = 7) and W^sh^^−/−^ (*n* = 7) littermate and mMCP4^−/−^ (*n* = 11) and mMCP4^+/−, +/−^ (*n* = 25) littermate mice were used for the study. Statistical differences in (**a**,**c**) were evaluated by two-way ANOVA with Sidak’s multiple comparison test. * *p* < 0.05, ** *p* < 0.01 Wsh^−/−^ vs. Wsh^+/−^ (MC903-treated), ### *p* < 0.001 vs. vehicle-treated, ns, not significant.

**Figure 2 ijms-21-06311-f002:**
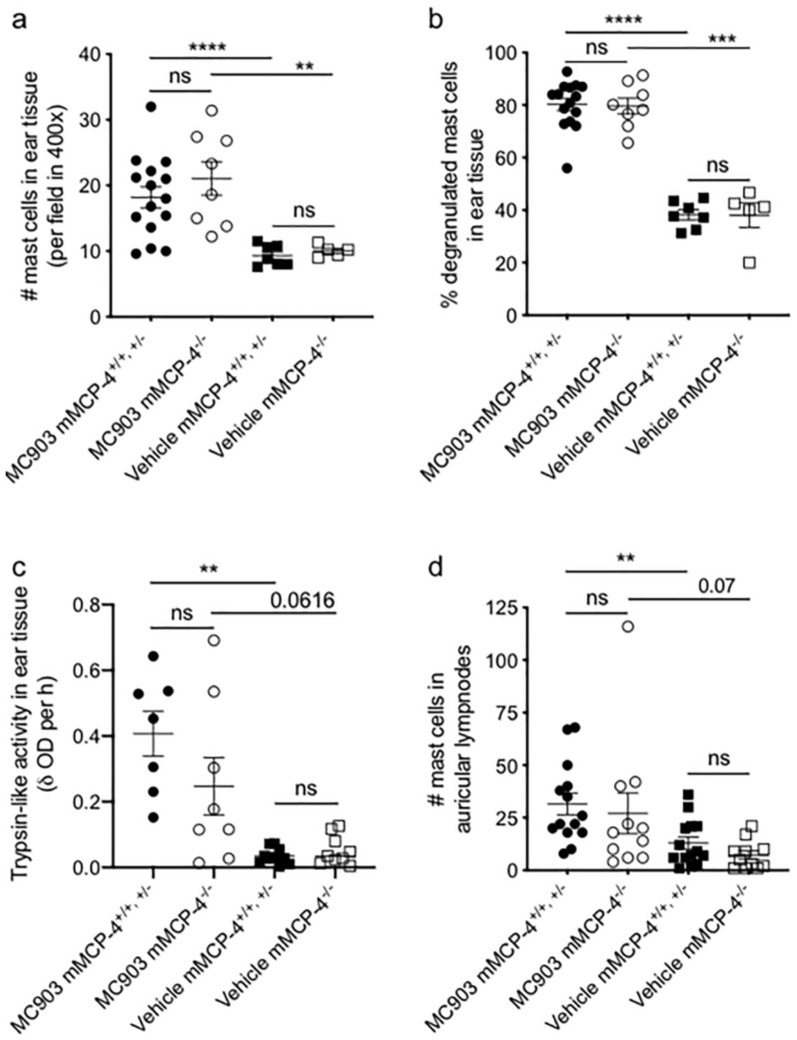
Mast cell recruitment and activation in MC903-induced atopic dermatitis are not impeded in absence of the mast cell chymase mMCP-4. Topical application of MC903 for 14 days resulted in increased (**a**) mast cell infiltration, (**b**) mast cell degranulation and (**c**) trypsin-like activity, in the ear tissue of both mMCP4^−/−^ and mMCP4^+/+, +/−^ mice compared to vehicle treatment; (**d**) mast cell infiltration in the auricular lymph nodes draining the MC903-treated ears of both mMCP4^−/−^ and mMCP4^+/+,+/−^ mice compared to the auricular lymph nodes draining vehicle-treated ears. Age-matched 6–9 weeks old littermate mice were used for the study. ** *p* < 0.01, *** *p* < 0.001 and **** *p* < 0.0001, ns, not significant.

**Figure 3 ijms-21-06311-f003:**
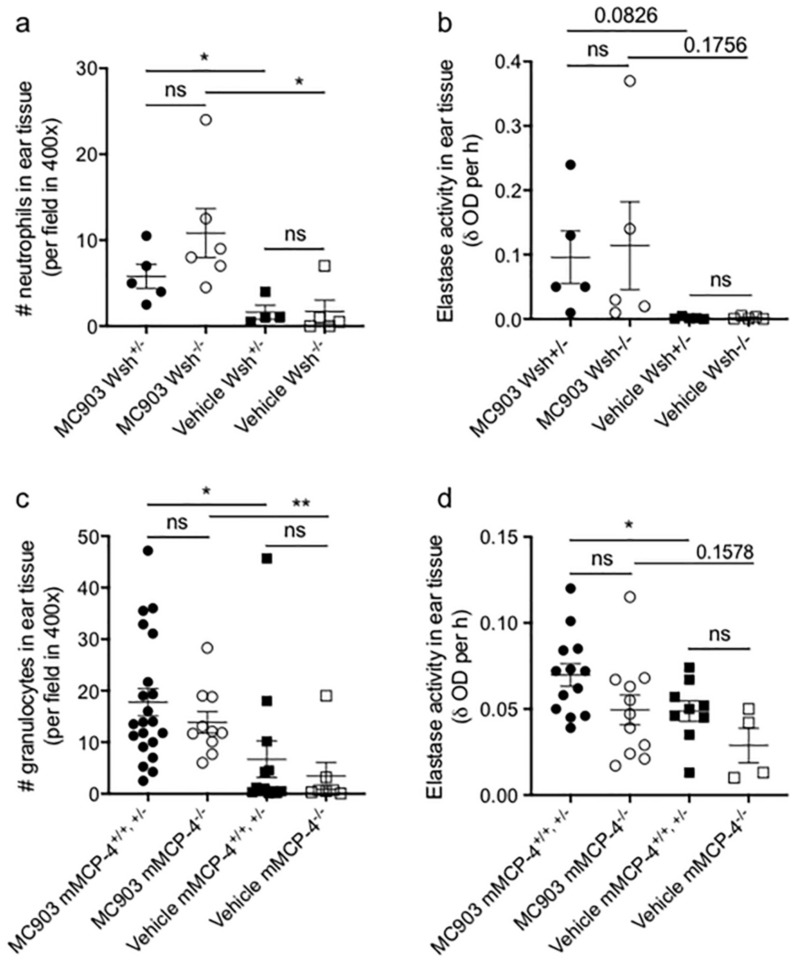
Significant influx of granulocytes to the ear tissue in MC903-induced atopic dermatitis. (**a**,**c**) neutrophil or granulocyte infiltration and (**b**,**d**) neutrophil elastase (NE) activity in MC903-treated Wsh^+/−^ and Wsh^−/−^ ears (**a**,**b**) and mMCP4^−/−^ and mMCP4^+/+,+/−^ ears (**c**,**d**) compared to vehicle treated ears. Age matched 6–9 weeks old littermate mice were used for the study. Number of mice analyzed: in (**a**) MC903-treated Wsh^+/−^ (*n* = 5), Wsh^−/−^ (*n* = 6), vehicle treated Wsh^+/−^ (*n* = 4), Wsh^−/−^ (*n* = 5); in (**b**) MC903-treated Wsh^+/−^ (*n* = 5), Wsh^−/−^ (*n* = 5), vehicle treated Wsh^+/−^ (*n* = 5), Wsh^−/−^ (*n* = 5); in (**c**) MC903-treated mMCP4^+/+, +/−^ (*n* = 21) and mMCP4^−/−^ (*n* = 11), vehicle treated mMCP4^+/+, +/−^ (*n* = 12) and mMCP4^−/−^ (*n* = 7); and in (**d**) MC903-treated mMCP4^+/+, +/−^ (*n* = 13) and mMCP4^−/−^ (*n* = 11), vehicle treated mMCP4^+/+, +/−^ (*n* = 9) and mMCP4^−/−^ (*n* = 4). * *p* < 0.05, ** *p* < 0.01 and ns, not significant.

**Figure 4 ijms-21-06311-f004:**
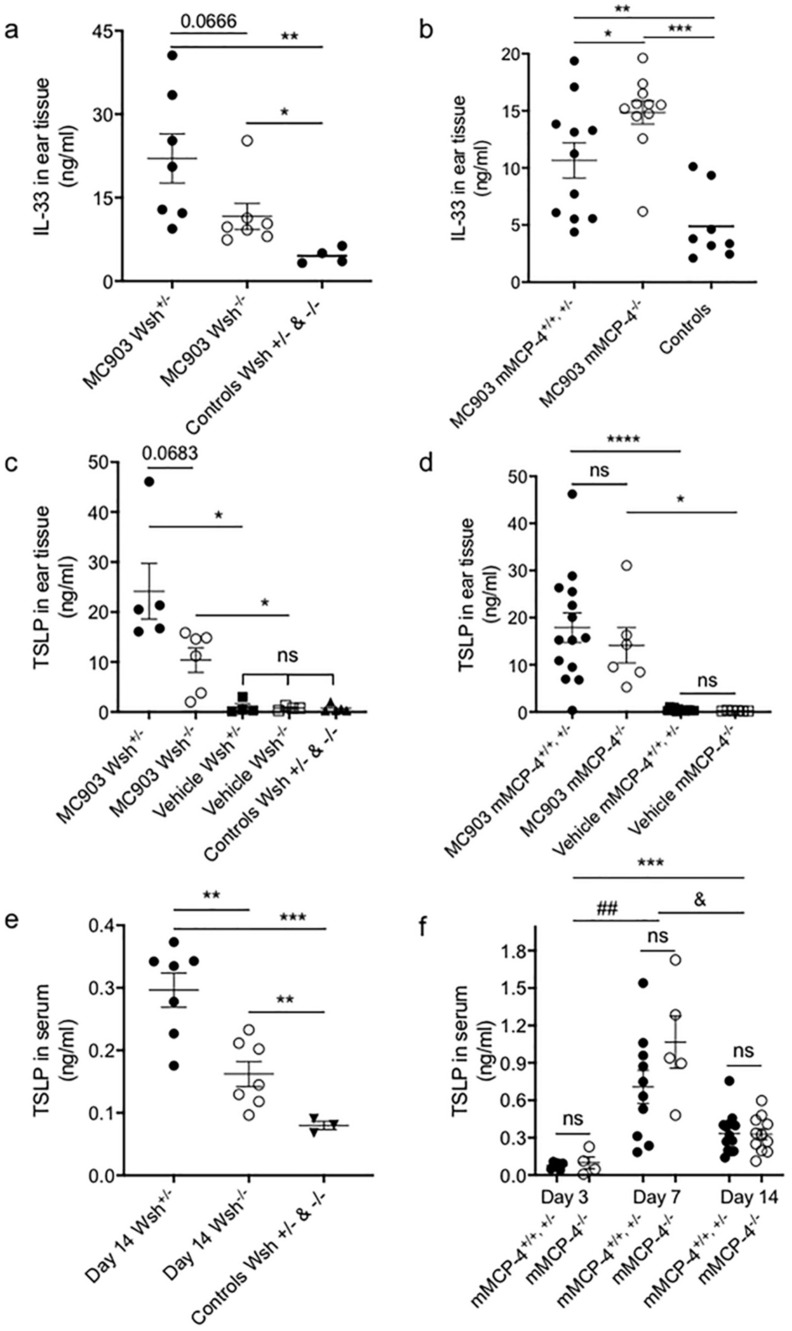
Treatment with MC903 increases production of IL-33 and thymic stromal lymphopoetin (TSLP). In age matched 6–9 weeks old littermate mice, the MC903 treatment (**a**) decreased the ear tissue levels of IL-33 level in the Wsh^−/−^ mice and (**b**) significantly increased IL-33 levels in the mMCP-4^−/−^ mice as compared to the mMCP-4^+/+, +/−^ mice. MC903 treatment increased TSLP levels in ear tissue (**c**,**d**) and in serum (**e**,**f**) as compared to vehicle treated Wsh^+/−^ and mMCP-4^+/+, +/−^ littermate mice and non-treated littermate controls. Note that c-kit deficiency reduces the MC903-induced TSLP levels. In figure panels (**a**–**e**) * *p* < 0.05, ** *p* < 0.01, *** *p* < 0.001 and **** *p* < 0.0001, and in figure panel (**f**) ^##^
*p* < 0.01 vs. day 3, ^&^
*p* < 0.05 vs. day 7 and *** *p* < 0.001 vs. day 3, ns, non-significant.

**Figure 5 ijms-21-06311-f005:**
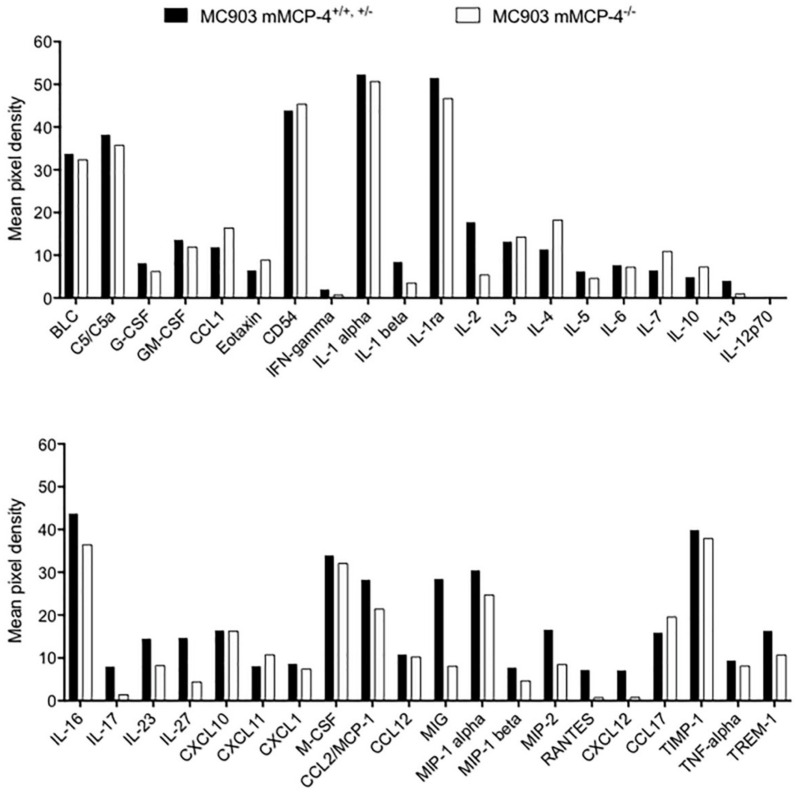
MC903-induced atopic dermatitis evokes the local production of inflammatory cytokines. Membrane cytokine array of ear tissue lysates from MC903-treated mMCP4^−/−^ and mMCP4^+/+, +/−^ ears of the age-matched 6–9 weeks old littermate mice demonstrate a massive production of inflammatory mediators. Note the almost complete absence of the Th1-type cytokines, IL-12 and IFN-γ.

**Figure 6 ijms-21-06311-f006:**
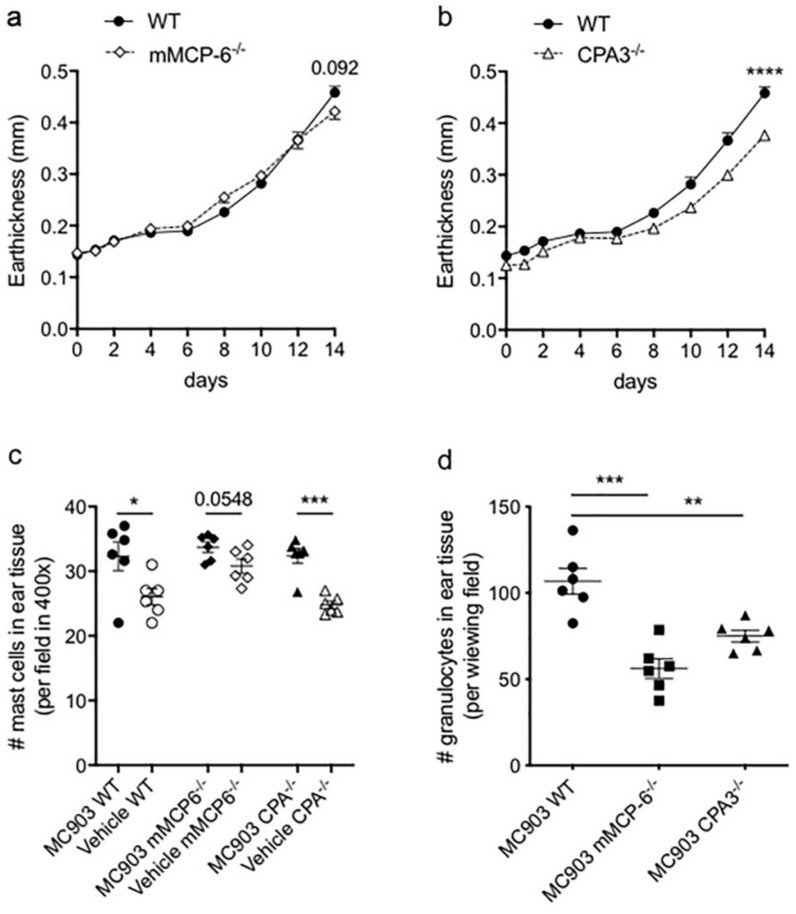
MC903-induced clinical symptoms and leukocyte infiltration in mMCP-6- and CPA3- deficient mice. Significant increase in ear thickness in (**a**) mMCP-6^+/+^ and mMCP-6^−/−^ and (**b**) CPA3^+/+^ and CPA3^−/−^ littermate mice, compared to vehicle treated ears. MC903-induced (**c**) mast cell and (**d**) granulocyte infiltration, as compared to vehicle treated ears. * *p* < 0.05, ** *p* < 0.01, *** *p* < 0.001 and **** *p* < 0.0001, ns, non-significant.

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
