# Peer review of "Mast Cells Limit Ear Swelling Independently of the Chymase Mouse Mast Cell Protease 4 in an MC903-Induced Atopic Dermatitis-Like Mouse Model"

_ijms, 2020, doi:10.3390/ijms21176311_

Round 1

Reviewer 1 Report

The authors investigated the role of mast cells (MC) and MC specific proteases on the pathogenesis of atopic dermatitis (AD). They used a vitamin D3-analog (MC903) to induce clinical AD-like symptoms in MC-deficient (Wsh-/-) and the MC protease-deficient (mMCP-4 -/-) , mMCP-6 -/- , and CPA3 -/- mouse strains. The authors showed that MC could modulate MC-proteases mediated inflammation. They found upregulation of both TSLP and IL33 relevant regulators of AD in mice applied with MC903 compared to the vehicle but did not find the role of mMCP4 in regulating TSLP. Overall, the authors demonstrated that MCs are essential in AD-like inflammation, clinical, and systemic symptoms. The authors found that mMCP4 plays a significant role in regulating the bioavailability of IL33. While the study is interesting, but the way the results are presented, it is confusing.  Please see my comments below for more details.

General comments:

1). The authors presented all the non-significant data in the first three figures and some of the important data were presented in supplemental figures so that it is reported as major concerns.

2). The authors didn’t run the statistics on a comparison between control and mutant mice from mMCP4, which in the reviewer’s opinion, should be important to know for most of the figures, including the supplementary figures where comparison was not made between control and mutant mice with MC903 application. To note in some cases, it is clear that the difference was not significant, but it is still essential to run the statistics and should be presented. Another concern, line 29-30, is presented as a significant finding and conclusion in the abstract, but the data were shown as supplementary figures.

3). As shown in this paper (PMID: 22901752), that mouse chymase mMCP4 degrades TNF, limits inflammation, and promotes survival in a model of sepsis. Although it is a different model, the role of mMCP4 is involved in the degradation of TNA-alpha, however, in this study, as illustrated in Figure 4, authors did not find any difference in TNF-alpha in their result. How would the authors explain this difference?

4). Figure 5C, authors, showed an increasing trend of TSLP in serum, but at day 13, it comes to baseline. It is unclear if an increase in TSLP from day 2 to day 6 is significant, and then on day 13, authors found the serum availability of TSLP is reduced to baseline. This should be discussed in the discussion.

5). Some of the statements were cited with incorrect references. For instance, line 37 and line 40, these citations are not relevant to humans.

6). Scientific rigor is an important issue; to that aspect, the authors did not mention how did they analyze their study?

7). Supplementary figure 3, Y-axis for 5 nmol data is unclear the way it is shown in the figure.

8). In general, no explanation was provided in the method section on how the field was selected for the estimation of different immune cell types counting.

9). Supplementary figure 6d, it was unclear what is per viewing field, and also, the genetic background of mice.

Reviewer 2 Report

The original research paper evaluated here describes the role of mast cells (MCs) in atopic dermatitis (AD)-like inflammation using murine model of this disease and several mutant strains. To induce AD phenotype in a mouse, topical low-calcemic vitamin D analogue (MC903) was used. Authors concluded that MCs both promote and control the level of the MC903-induced AD-like inflammation.

The article is suitable for publication subject to major revisions according to the following critiques:

  1. Line 155/259: I have concerns regarding classification of TNF-alpha to Th2-like cytokines in the manuscript. I think that this cytokine belongs to Th1-like cytokines. Please provide appropriate citations or classified TNF-alpha to Th1.
  2. Line 188: “Serum levels day 2 post-treatment…” This sentence is not clear. Serum level refers to TSLP? Please specify.
  3. Line 216: I can read: “However, several experimental allergic models have suggested that the effective impact of MCs can be overruled by a to high dose”. Again, please be more precise. … by a to high dose of this vitamin? Anyway, as you propose to use lower dose of MC903 to induce AD-like phenotype, please provide an appropriate skin pictures and prove that 8-times lower MC903 dose (compared to Li et al. 2006) can also induce the same or at least similar AD clinical phenotype. It can be presented in a supplementary data.
  4. The quality of Supplementary Figure 1 is very low. Please provide a picture with a better resolution.
  5. Figure legends. Please avoid expression: *=P<0.05. Please delete “=”.

Finally, because authors used MCs-deficient mice (or lacking the MCs-specific enzymes), it would be interesting to know the impact of topical MC903 treatment on AD phenotype and the association with IgE. Did authors measure the level of IgE in those mice strains and noticed any differences between WT and MCs-deficient mice (+/- MC903)? Please discuss the role of IgE in regard to the outcomes of the manuscript, especially in the context of MCs-deficient mice. Any associations?

Round 2

Reviewer 1 Report

Authors addressed all the concerns!

Reviewer 2 Report

Thank you for answering my comments. I have no more comments.